# The CIPCA-BPNN Failure Prediction Method Based on Interval Data Compression and Dimension Reduction

**Linchao Yang [1], Guozhu Jia [1], Fajie Wei [1], Wenbing Chang [2], Chen Li [3] and Shenghan Zhou [2],***

[1]  School of Economics and Management, Beihang University, Beijing 100191, China; yanglinchao@buaa.edu.cn (L.Y.); jiaguozhu@buaa.edu.cn (G.J.); weifajie@buaa.edu.cn (F.W.)
[2]  School of Reliability and Systems Engineering, Beihang University, Beijing 100191, China; changwenbing@buaa.edu.cn
[3]  Poly Huixin Investment Co., Ltd., Beijing 100010, China; xinx_yang@buaa.edu.cn
[*]  Correspondence: zhoush@buaa.edu.cn

**Abstract:** This paper proposes a complete-information-based principal component analysis (CIPCA)-back-propagation neural network (BPNN)_ fault prediction method using real unmanned aerial vehicle (UAV) flight data. Unmanned aerial vehicles are widely used in commercial and industrial fields. With the development of UAV technology, it is imperative to diagnose and predict UAV faults and improve their safety and reliability. The data-driven fault prediction method provides a basis for UAV fault prediction. A UAV is a typical complex system. Its flight data is a kind of typical high-dimensional large sample dataset, and traditional methods cannot meet the requirements of data compression and dimensionality reduction at the same time. The method used interval data to compress UAV flight data, used CIPCA to reduce the dimensionality of the compressed data, and then used a back propagation (BP) neural network to predict UAV failure. Experimental results show that the CIPCA-BPNN method had obvious advantages over the traditional principal component analysis (PCA)-BPNN method and could accurately predict a failure about 9 s before the UAV failure occurred.

**Keywords:** fault diagnostics and prognostics; UAV flight data; interval data; complete-information-based principal component analysis; BP neural network

## 1. Introduction

Unmanned aerial vehicles (UAVs) are very versatile and can be used in personal and commercial fields such as aerial photography, agriculture, plant protection, miniature selfies, express transportation, disaster relief, surveying and mapping, and electric power inspection. In order to reduce costs, UAVs usually adopt non-redundant or low-redundancy design. In addition, due to the lack of driver's real-time observation and judgment ability during flight, UAVs have a high accident rate. Improving the safety and reliability of equipment has become a research hotspot [1]. The traditional UAV fault prediction approach is to monitor a certain flight parameter, and when the parameter exceeds the safe range, or when it is judged that it may exceed the safe range in the future, a risk alarm is issued [2–4]. However, UAVs are complex systems, and sometimes it is difficult to locate fault variables. In addition, due to the small number of fault samples, it is difficult to support the establishment of mathematical models of variables to predict the next state of risk variables [5]. Some scholars use the image returned by the UAV's own camera to locate and estimate the bounded domain of the UAV's attitude and to perform fault detection based on the landmark error of the UAV's tracking image [6]. However, none of these methods use all of the flight information. With the maturity of machine learning methods, data-driven fault diagnosis methods have become a hot research topic, and there are a lot of research studies and applications in many fields, such as bearings [7–9], power distribution networks [10], and photo-voltaic array fault diagnosis [11]. The data-driven

method uses all the information that the UAV system can collect. After using the machine learning method, it does not rely on the original model of the system to judge the fault [12]. The construction of data-driven methods usually includes three steps: first, collecting fault signals; second, extracting fault features; third, identifying and predicting fault [13]. Because the airborne equipment of UAVs record a large amount of flight parameter data in real time, feature extraction and dimensionality reduction of the flight data are very important tasks.

Flight data are real-time flight data collected by onboard sensors, which are a type of signal data. Traditional signal data feature extraction methods including wavelet packet transform (WPT), empirical mode decomposition (EMD), and local mean decomposition (LMD). WPT is a signal feature extraction method that provides local features in the time domain and frequency domain and recognizes sudden components of vibration signals. It is an effective method for processing nonlinear and non-stationary signals [14,15]. EMD is an adaptive processing method suitable for analyzing nonlinear and non-stationary signals. The algorithm is based on the local characteristic time scale of the signal and has the ability to adaptively decompose the complex signal into multiple independent modal functions [16]. LMD is also an adaptive signal processing method used to adaptively de-compose nonlinear and non-stationary vibration signals into a series of product functions [17]. WPT needs to determine the decomposition scale, so it is not an adaptive signal data processing method and is not conducive to processing big data [18]. EMD can adaptively determine the resolution of the signal in different frequency bands, but the modal mixing problem often occurs [19]. LMD and EMD have some similarities, but LMD is better than EMD in the processing of local signal features [20]. These methods are often used in the field of fault diagnosis [21–24], but they have some problems. First, extracting data features from the time and frequency domains will destroy the structure of the data itself. Second, these methods will increase the number of variables and cannot achieve the purpose of dimensionality reduction.

Principal component analysis based on interval data can solve the problems of compressed data, feature extraction, and dimensionality reduction at the same time. In 1988, Diday proposed symbolic data analysis (SDA), which has been widely used in various fields [25]. Interval data are typical symbolic data, which express a range between the upper and lower bounds. Compared with discrete data, interval data can grasp the internal structural characteristics of data objects globally, which is more conducive to revealing the rules implicit in the data. Therefore, interval data can represent the uncertainty and variability of data and have important application value in decision support. Wang, Guan, and Wu (2012) proposed the complete-information-based principal component analysis (CIPCA), which can capture the complete information of the data interval and find the meaningful structural information hidden in large-scale data. It is a more efficient method for dimensionality reduction of large-scale numerical data [26]. The interval data principal component method can distinguish fault types more accurately than the traditional principal component analysis method [27]. The principal component analysis method of interval data has been widely used in sensor fault diagnosis [28–30], spacecraft fault diagnosis [31], and other fault diagnosis fields. This paper introduces CIPCA into the UAV fault prediction and uses a back-propagation neural network (BPNN) to construct the CIPCA-BPNN fault prediction model. For the purpose of failure prediction, we used real labeled UAV flight data and selected flight data 30 s before the fault as fault data.

The rest of this paper is organized as follows. Section 2 describes the interval data and CIPCA. Section 3 discusses the application process of CIPCA in UAV failure prediction. Section 4 describes the experiment in this paper. Section 5 analyzes the experimental results. In Section 6, the conclusions are given.

## 2. Methods

### 2.1. Interval Data

Interval data refers to the idea that the feature of a sample point is not a definite value but is a collection of all values contained in a range on the real number field, which can be expressed as

$$x = \{t | \underline{x} \leq x \leq \overline{x}, \underline{x} \in R, \overline{x} \in R, \underline{x} \leq \overline{x}\} \tag{1}$$

where $\underline{x}$ is the lower bound of interval data, and $\overline{x}$ is the upper bound of interval data. Interval data express a range between the upper and lower bounds. Compared with discrete data, interval data can summarize the internal structural characteristics of the data from a global perspective, and they are more conducive to explaining the rules implicit in the data. Interval data can also be represented as an ordered array of upper and lower bounds: $x = [\underline{x}, \overline{x}]$.

For an n-dimensional vector $X = (x_1, x_2, \ldots, x_n)^T$, if each component in the vector is interval data, that is, $x_i = [\underline{x_i}, \overline{x_i}]$, then $X$ is called an n-dimensional interval vector. If each datum in the $n \times p$ dimensional data matrix $X_{n \times p} = (x_{ij})_{n \times p}$ is an interval datum, it is called interval matrix:

$$X_{n \times p} = (x_{ij})_{n \times p} = \begin{pmatrix} e_1^T \\ e_2^T \\ \vdots \\ e_n^T \end{pmatrix} = (X_1, X_2, \ldots, X_p) \tag{2}$$

Each row in the matrix is an interval sample, and the number of columns $p$ represents the sample dimension. In fault diagnosis, the interval matrix can be used to describe the data, and the observation value of each sample dimension is represented by a data interval.

### 2.2. CIPCA

The research object of interval data principal component analysis is an interval data matrix $X_{n \times p}$ containing $n$ samples; each sample is described by $p$ interval variables:

$$\begin{aligned} X_{n \times p} \quad &= (X_1, X_2, \ldots, X_p) \\ &= \begin{bmatrix} [\underline{x_{11}}, \overline{x_{11}}] & [\underline{x_{12}}, \overline{x_{12}}] & \cdots & [\underline{x_{1p}}, \overline{x_{1p}}] \\ [\underline{x_{21}}, \overline{x_{21}}] & [\underline{x_{22}}, \overline{x_{22}}] & \cdots & [\underline{x_{2p}}, \overline{x_{2p}}] \\ \vdots & \vdots & \ddots & \vdots \\ [\underline{x_{n1}}, \overline{x_{n1}}] & [\underline{x_{n2}}, \overline{x_{n2}}] & \cdots & [\underline{x_{np}}, \overline{x_{np}}] \end{bmatrix} \end{aligned} \tag{3}$$

Many principal component analysis methods for interval data have been proposed. Cazes, Chouakria, and Diday (1997) proposed vertices principal component analysis (VPCA) and centers principal component analysis (CPCA) [32]. However, these two methods have the disadvantage of using only local information. Wang, Guan, and Wu (2012) proposed the complete-information-based principal component analysis for interval data (CIPCA) [26]. This method uses all the information of the interval samples. The modeling results always reflect the internal structural characteristics of the data and are not easily affected by the size of the interval samples. Compared with VPCA and CPVA, CIPCA has higher accuracy and stronger robustness.

The same as traditional principal component analysis, in CIPCA, the $k$-th interval principal component $P_k$ is a linear combination of $p$ interval variables, i.e., $P_k = u_{1k}X_1 + u_{2k}X_2 + \cdots + x_{pk}X_p$, where $u_k = (u_{1k}, u_{2k}, \ldots, u_{pk})\prime \in R^p$ subject to $u_k'u_k = 1$, and $u_k'u_l = $

0 $(1 \leq l, k \leq p, l \neq k)$. Using variance to describe the information contained in the principal component of the $k$-th interval, we have

$$
\begin{aligned}
D_{CI}(P_k) &= \frac{1}{n} \langle P_k, P_k \rangle \\
&= \frac{1}{n} \langle u_{1k}X_1 + u_{2k}X_2 + \cdots + u_{pk}X_p, \ u_{1k}X_1 + u_{2k}X_2 + \cdots + u_{pk}X_p \rangle \\
&= \frac{1}{n}(u_{1k}, u_{2k}, \ldots, u_{pk})
\begin{pmatrix}
\langle X_1, X_1 \rangle & \langle X_1, X_2 \rangle & \cdots & \langle X_1, X_p \rangle \\
\langle X_2, X_1 \rangle & \langle X_2, X_2 \rangle & \cdots & \langle X_2, X_p \rangle \\
\vdots & \vdots & \ddots & \vdots \\
\langle X_p, X_1 \rangle & \langle X_p, X_2 \rangle & \cdots & \langle X_p, X_p \rangle
\end{pmatrix}
\begin{pmatrix}
u_{1k} \\
u_{2k} \\
\vdots \\
u_{pk}
\end{pmatrix}
\end{aligned}
\tag{4}
$$

According to the principal component analysis, the sum of the variances of the first m principal components $P_1, P_2, \ldots, P_m$ should reach the maximum, so $m$ standard orthogonal vectors $u_1, u_2 \ldots, u_m$ should be solved to maximize $\sum_{k=1}^{m} D_{CI}(P_k)$ and satisfy $D_{CI}(P_1) \geq D_{CI}(P_2) \geq \cdots \geq D_{CI}(P_m)$ at the same time; it can be expressed as

$$
s.t. \begin{cases}
\max \sum_{k=1}^{m} u'_k S^{CI} u_k \\
u'_k u_k = 1 \\
u'_k u_l = 0 \\
u'_1 S^{CI} u_1 \geq u'_2 S^{CI} u_2 \geq \cdots \geq u'_m S^{CI} u_m \\
l = 1, 2, \ldots, m \ (l \neq k)
\end{cases}
\tag{5}
$$

The modeling steps of CIPCA are as follows:

Step 1: Normalize all interval variables to obtain the standardized interval data matrix $X_{n \times p}^*$. The normalization method is as follows:

$$
x_{ij}^* = \left[ \frac{x_{ij} - E_{CI}(X_j)}{D_{CI}(X_j)}, \ \frac{\overline{x_{ij}} - E_{CI}(X_j)}{D_{CI}(X_j)} \right]
\tag{6}
$$

Step 2: Calculate the covariance matrix $S^{CI}$ of $X_{n \times p}^*$.

Step 3: Perform feature decomposition on $S^{CI}$ to obtain eigenvalues $\lambda_1, \lambda_2, \ldots, \lambda_p$ ($\lambda_1 \geq \lambda_2 \geq \cdots \geq \lambda_p$) and corresponding standard orthogonal eigenvectors $u_1, u_2, \ldots, u_p$, and retain the first $m$ ($m \leq p$) eigenvalues and eigenvectors. Record principal component variance and principal component coefficient.

Step 4: Calculate the principal component score $P_1, P_2, \ldots, P_m$ of the interval.

### 2.3. BPNN

Artificial neural networks are supervised machine learning methods, which have been applied in many fields [33,34]. In the field of machine learning, back-propagation (BP) is a classical method used to train neural networks [35,36], which can deal with complex nonlinear system problems and is widely used in the field of fault diagnosis and prediction. A three-layer BPNN is shown in Figure 1. The first layer is the input layer of the BP neural network; assuming there are n variables, the input vector $x \in R^n$, where $x = (x_0, x_1, \ldots, x_{n-1})^T$. The second layer is the hidden layer, with a total of $l$ neurons, and its output is $h \in R^l$, $h = (h_0, h_1, \ldots, h_{l-1})^T$. The last layer is the output layer $y \in R^m$, $y = (y_0, y_1, \ldots, y_{m-1})^T$. $w_{ij}$ is the weight of the $i$-th neuron in the input layer to the $j$-th neuron in the hidden layer, and the threshold of the $j$-th neuron in the hidden layer is $\theta_j$. $u^{jk}$ is the weight of the $j$-th neuron in the hidden layer to the $k$-th neuron in the output

layer, and the threshold of the $k$-th neuron in the output layer is $\eta_j$. The mapping from the input layer to the hidden layer to the output layer can be expressed as

$$
\begin{cases}
h_j = f\left(\sum\limits_{i}^{n-1} w_{ij}x_i - \theta_j\right),\ j = 0, 1, \ldots, l-1 \\
y_k = f\left(\sum\limits_{j}^{l-1} u_{jk}z_j - \eta_k\right),\ k = 0, 1, \ldots, m-1
\end{cases}
\tag{7}
$$

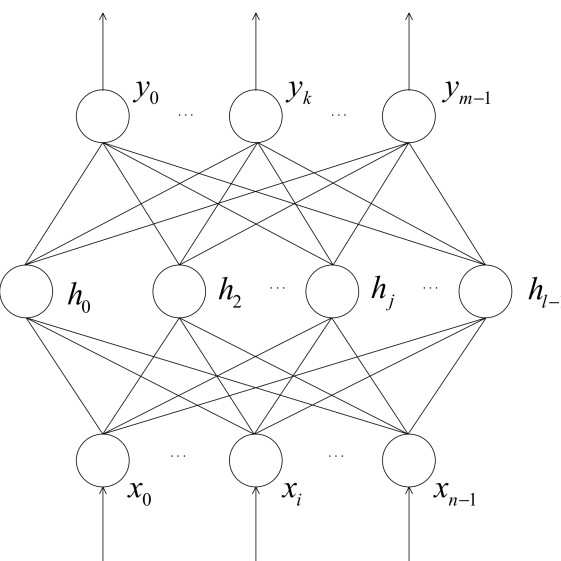

**Figure 1.** The structure of three-layer neural network.

The key to a BP neural network is to learn the weights and thresholds of the network through samples. The learning process is composed of the forward propagation of signals and the backward propagation of errors. Forward propagation means that after passing through the input layer and the hidden layer, the input signal is output to the output layer; if the desired output signal cannot be obtained, it is transferred to the reverse propagation of the error signal. In the back propagation of the error signal, the error signal is fed back layer by layer from the output layer, and each weight is adjusted by the error feedback, and through this continuous correction, the network output is closer to the expected output.

## 3. Application

### 3.1. Flight Data of UAV

The flying parameter system of a UAV records the entire process data from the start to the stop of the UAV, including attitude, altitude, power, navigation parameters, and other indicators. With the development of machine learning technology, data-driven diagnosis technology has become an important part of fault diagnosis. For UAVs, flight data are the basis for fault diagnosis and prediction. But in the actual flight data, there are two problems. First, the flying data set is huge. Airborne equipment usually records data in milliseconds, and the number of samples recorded during a flight may reach hundreds of thousands. Therefore, before using the flight data for analysis, the data should be compressed at the variable level. Second, there are a large number of flight status indicators in flight data, ranging from dozens to hundreds. The relationship between indicators is complicated, and the correlation is serious. Therefore, it is necessary to reduce the dimensionality of the data before using the flying parameter data for modeling. Using the idea of interval data and CIPCA method can solve these two problems well.

### 3.2. Compression Based on Flight Data

Due to the huge dimensions of flight data samples, if directly used for modeling, they will increase the computer load and greatly decrease the efficiency of modeling. In addition, the millisecond data collected by the sensors will obscure the information contained in the flight process. Therefore, before fault diagnosis modeling, the sample dimension of the flying parameter data should be compressed. The original data can be packaged using the idea of interval data, retaining the maximum and minimum values of various variables within a sample period of time, grasping the intrinsic characteristics of data objects globally, and compressing massive data. For data with time labels, interval length needs to be determined from the time dimension to ensure the timing of data and ensure that the time represented by each data interval is equal. For data without a time label, interval can be performed by other characteristics of the data. It is important to ensure that each interval has the same meaning in at least one feature of the data.

$T_{m \times p}$ is the interval data matrix after compressing the original data; the compression method can be expressed as

$$
\begin{aligned}
T_{m \times p} &= (T_1, T_2, \ldots, T_p) \\
&= \begin{pmatrix}
\left[ t_{11}, \overline{t_{11}} \right] & \left[ t_{12}, \overline{t_{12}} \right] & \cdots & \left[ t_{1p}, \overline{t_{1p}} \right] \\
\left[ t_{21}, \overline{t_{21}} \right] & \left[ t_{22}, \overline{t_{22}} \right] & \cdots & \left[ t_{2p}, \overline{t_{2p}} \right] \\
\vdots & \vdots & \ddots & \vdots \\
\left[ t_{m1}, \overline{t_{m1}} \right] & \left[ t_{m2}, \overline{t_{m2}} \right] & \cdots & \left[ t_{mp}, \overline{t_{mp}} \right]
\end{pmatrix}
\end{aligned}
\tag{8}
$$

where $t_{ij} = \min\left( \left[ x_{l,j}, x_{l+1,j}, \ldots, x_{l+k,j} \right] \right)$, and $\overline{t_{ij}} = \max\left( \left[ x_{l,j}, x_{l+1,j}, \ldots, x_{l+k,j} \right] \right)$. The length of time included in $t_{ij}$ and $\overline{t_{ij}}$ is the time period spanned by $x_{l,j}, x_{l+1,j}, \ldots, x_{l+k,j}$.

After interval compression, the sample size of the original data can be greatly reduced, and using the interval feature that retains the original data, less information is lost, which is beneficial for subsequent modeling and analysis of failures.

### 3.3. Dimensional Reduction of Flight Data Based on CIPCA

Because the interval matrix $T_{m \times p}$ of the original data is obtained based on the interval compression of samples, the dimension reduction method for interval data is needed when reducing variable dimensions. According to the CIPCA in Section 2.2, first, standardize the interval variable $(T_1, T_2, \ldots, T_p)$ and calculate the covariance matrix of $T_{m \times p}$ to obtain the eigenvalues and the corresponding standard orthogonal eigenvectors, retaining the first $q$ eigenvalues and eigenvectors. The principal component score $P_{m \times q}$ of the interval data is calculated. $P_{m \times q}$ is the interval type flight data, $X_{n \times p}$, obtained by the original massive high-dimensional flying parameter data after sample size compression and variable dimension reduction.

### 3.4. Fault Prediction of UAV Based on CIPCA-BP

Based on the interval matrix $P_{m \times q}$ obtained above, a UAV fault diagnosis model can be established through BPNN. From the interval matrix $P_{m \times q}$, the minimum and maximum values of the interval data are extracted and form the minimum matrix $P_{m \times q}^{\min}$

and a maximum matrix $P_{m \times q}^{\max}$, respectively. Since $p_{j*}$ and $\overline{p_{j*}}$ come from the same data interval, $P_{m \times q}^{\min}$ and $P_{m \times q}^{\max}$ have the same status label vector $y = (s_1, s_2, \ldots, s_m)^T$.

$$P_{m \times q}^{\min} = \begin{pmatrix} p_{11} & p_{12} & \cdots & p_{1q} \\ p_{21} & p_{22} & \cdots & p_{2q} \\ \vdots & \vdots & \ddots & \vdots \\ p_{m1} & p_{m1} & \cdots & p_{mq} \end{pmatrix} P_{m \times q}^{\max} = \begin{pmatrix} \overline{p_{11}} & \overline{p_{12}} & \cdots & \overline{p_{1q}} \\ \overline{p_{21}} & \overline{p_{22}} & \cdots & \overline{p_{2q}} \\ \vdots & \vdots & \ddots & \vdots \\ \overline{p_{m1}} & \overline{p_{m2}} & \cdots & \overline{p_{mq}} \end{pmatrix} \quad (9)$$

Using BPNN to establish fault prediction models for $\left(P_{m \times q}^{\min}, y\right)$ and $\left(P_{m \times q}^{\max}, y\right)$, respectively, the output layer uses linear functions to output the results; consequently, summarize the results of the two models to obtain the final prediction results. The process of CIPCA-BPNN is shown in Figure 2.

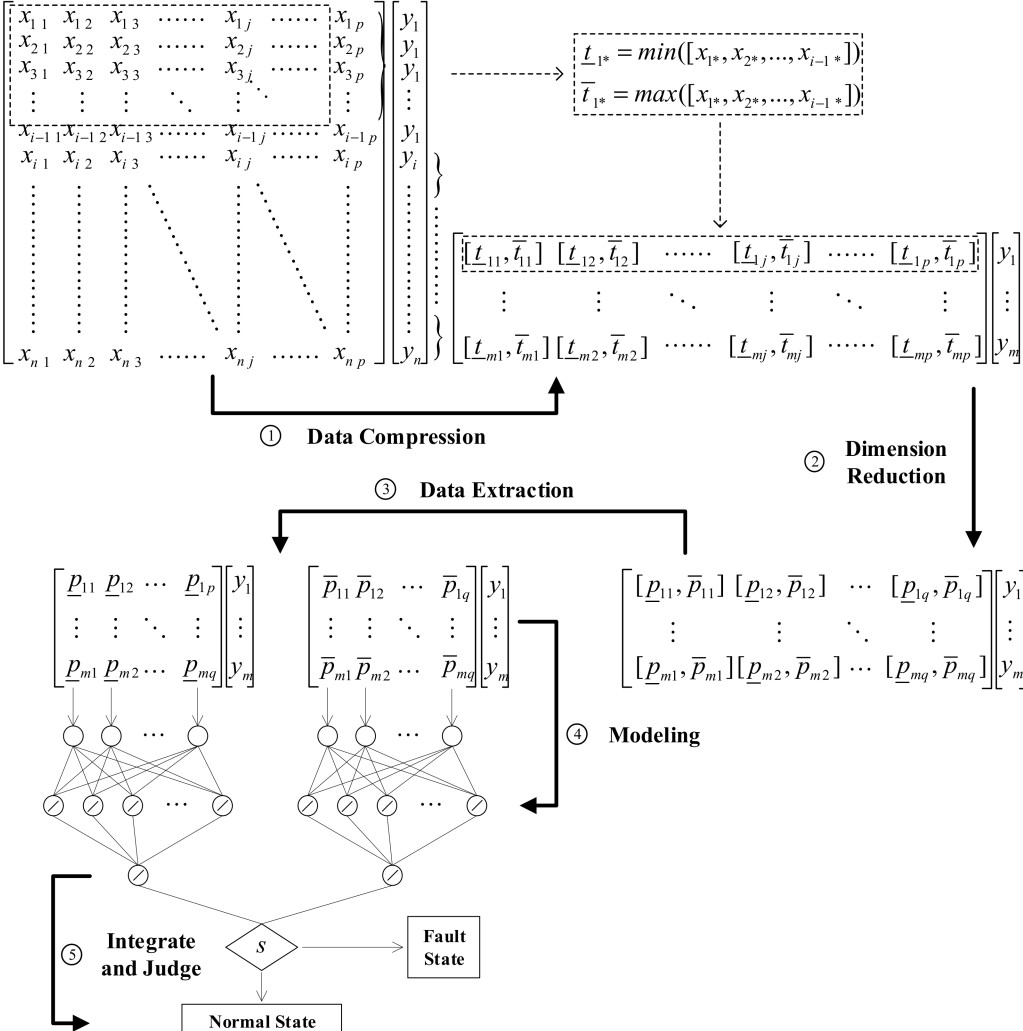

**Figure 2.** Complete-information-based principal component analysis–back-propagation neural network (CIPCA-BPNN) fault diagnosis process. The sample state of the same flight is the same ($y$ is the same). In the modeling stage, the data used are the data of many flights, and the $y$ of different states will be different.

## 4. Experiments Description

*4.1. Data*

The data in this paper came from the experimental data of the multi-rotor UAV VesperTilio of Volitation (Beijing) Technology Co., Ltd. (Beijing, China) The position markers of some airborne sensors are shown in Figure 3. In the study, the flight data of 123 sorties of a multi-rotor UAV were collected and collated. Among them, 10 sorties failed, and the other 113 sorties were normal flights. In order to realize the prediction of the fault, we extracted the data of 30 s before the fault occurred as the fault data. Similarly, in each normal flight, we also extracted continuous 30 s of data. There were a total of 16,471 fault sample points, 237,414 normal sample points, and 56 flight parameters. Due to the problem of sensor accuracy degradation and data loss in the fault state, the sampling frequency of the fault state and the normal state were different.

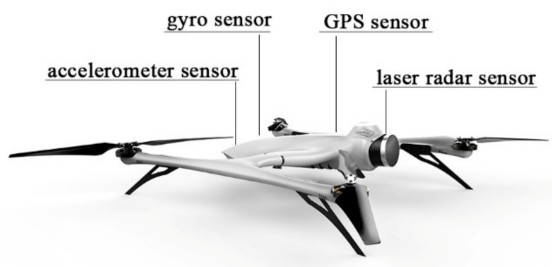

**Figure 3.** Airborne layout sensors (VesperTilio of Volitation (Beijing) Technology Co., Ltd.).

As can be seen from Figure 4, the fault data fluctuated more than normal data, and extreme fluctuations may have occurred suddenly. It was meaningless to use the data at each time point as a sample for fault prediction, because at such time points, the fault data may have been the same as the normal data. Therefore, we used the method of interval data for analysis, and could retain the fluctuation characteristics of some data. It can be seen from Figure 5 that the flight data variable had a large number of dimensions, and there was a serious correlation between the variables. Therefore, before modeling, we should have reduced the dimension of variables and removed the correlation between variables.

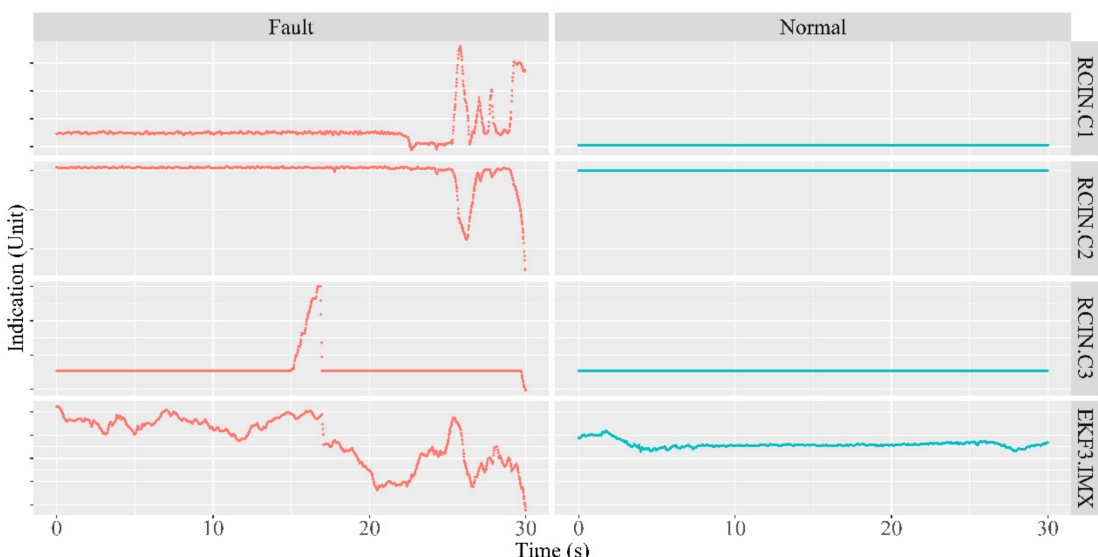

**Figure 4.** Comparison of fault data with normal data. The left side is the fault data, and the right side is the normal data. The fault data is not stable.

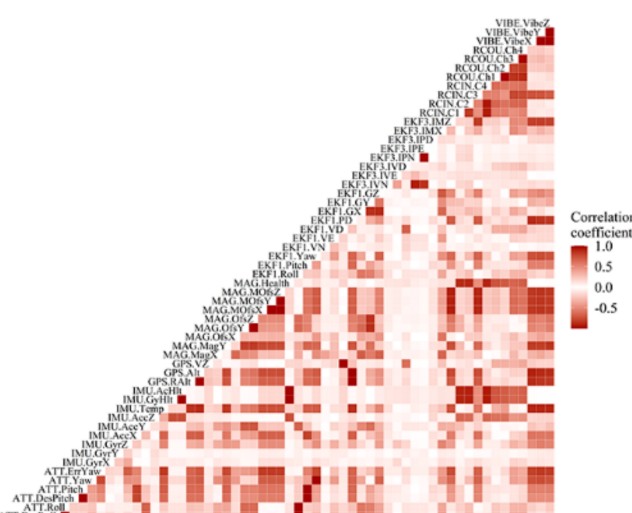

**Figure 5.** The correlation between variables of the data. The color is darker, the correlation between variables is higher. (The figure shows the correlation coefficient of the corresponding variable. Because the autocorrelation coefficient of the variable is 1, it is not shown in the figure.)

### 4.2. Experiments

#### 4.2.1. Data Compression

Interval data can achieve the purpose of compressing data sample size. For flight data, we can split data from the time dimension. In the experiment, the original flight data of each sorting was set as a data interval every 0.1 s, and the maximum and minimum values were extracted from the data interval to form the interval data matrix according to the method in Section 2.1. The samples of each sort were in the same state, so the interval processing of flight data did not affect the label of the samples (normal state or fault state). The obtained interval data was used for the subsequent dimension reduction based on CIPCA and the establishment of a fault prediction model based on BPNN.

#### 4.2.2. Dimension Reduction

We used CIPCA to carry out variable dimension reduction on the obtained flight interval data. The method is explained in Section 2.2. In addition, we compared the dimensionality reduction results of traditional PCA as a comparison. The data used by PCA were extracted from the original flight data of each sortie every 0.1 s, ensuring the consistency of the data volume with CIPCA.

#### 4.2.3. Fault Prediction

We used the interval data set obtained after dimensionality reduction by the CIPCA method to train the CIPCA-BPNN fault prediction model. Since the interval data could not be directly used to establish the BPNN model, according to the instructions in Section 3.4, we extracted the "minimum" matrix and the "maximum" matrix from the flight interval data set. We then used BPNN to train the "lower bound" base learner and the "upper bound" base learner, respectively, and summarized the results to form an "ensemble" learner. The method of summarizing the results was the weighted average.

In the failure prediction experiment, the label of the failure sample was 1 ($y = 1$), and the label of the normal sample was 0 ($y = 0$). We used two-thirds of all data as the training set, and the remaining one-third was the test set. We trained a single hidden layer BPNN with the number of hidden neurons of 10, 15, 20, 35, 30, 35, 40, and 45 to find the optimal model. The output function of BPNN was a linear function. We used the *nnet* package in *R* to build the BPNN model on a computer with the AMD Ryzen 7 1700 Eight-Core Processor 3 GHz CPU and 32 GB RAM. In the comparative experiment of PCA-BPNN, the same number of principal components as CIPCA-BPNN was used. The setting of PCA-BPNN parameters was the same as CIPCA-BPNN.

## 5. Results

### 5.1. Data Compression

The original data included 123 sorties of flight data, with a total of 253,885 sample points. We used the interval method to compress the original flight data of each sortie. After finishing, there were a total of 35,091 samples, including 2973 fault samples and 32,118 normal samples. After intervalization, the sample size was 13.82% of the original data. The effect of data compression is obvious. It can reduce the number of samples, retain the interval information of the original data, and alleviate the problem of data imbalance to a certain extent (Table 1).

**Table 1.** Effect of data compression and dimension reduction.

| Stage | Number of Samples | Number of Faults | Number of Normal | Imbalanced Ratio | Number of Variables |
|---|---|---|---|---|---|
| Raw data | 253,885 | 16,471 | 237,414 | 14.41 | 56 |
| Data compression | 35,091 | 2973 | 32,118 | 10.80 | 56 |
| Dimension Reduction | 35,091 | 2973 | 32,118 | 10.80 | 5 |

### 5.2. Dimension Reduction

After intervalization, the flight data were well compressed at the sample level. However, there were still a large number of variables, and the relationship between those variables was not clear. Besides, there may have been serious correlations. Therefore, before training the fault prediction model, the CIPCA in Section 2.2 was used to reduce the dimensionality of the intervalized flight data. In order to compare the effect of dimensionality reduction, we also used PCA to perform dimensionality reduction on the data. The data used by PCA are described in Section 4.

It can be seen from Figure 6 that the cumulative variance interpretation curve of CIPCA was located above the PCA, indicating that CIPCA had a stronger ability to interpret data than PCA and could better cover the information of the data. When all five principal components were extracted, CIPCA could explain more than 80% of the flight data, while PCA could only explain less than 40% of the information.

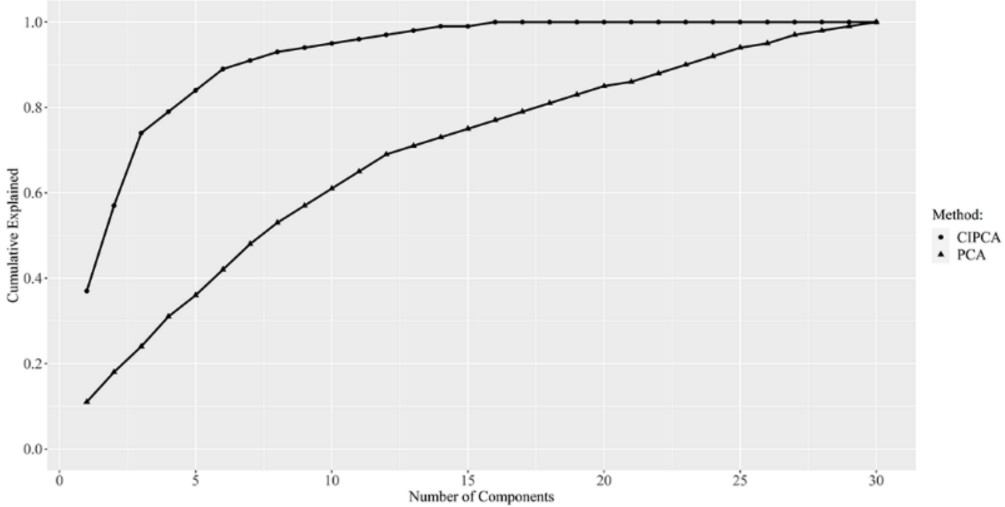

**Figure 6.** Comparison of CIPCA and principal component analysis (PCA) cumulative variance explained.

When retaining the number of principal components, we ensured that the retained principal components could explain most of the data information and chose as few principal component numbers as possible because too many principal components would have reduced the effect of dimensionality reduction. According to Table 2, the first five principal

components of CIPCA could explain nearly 85% of the data information, so we chose to retain the five to train BPNN for UAV failure prediction. Similarly, in order to compare the prediction effect, the first five principal components of PCA were also selected for modeling.

**Table 2.** Flight data variance explained by CIPCA and PCA.

| Number of Components | CIPCA | | PCA | |
|---|---|---|---|---|
| | Proportion Explained | Cumulative Explained | Proportion Explained | Cumulative Explained |
| 1 | 0.367 | 0.367 | 0.107 | 0.107 |
| 2 | 0.206 | 0.573 | 0.074 | 0.181 |
| 3 | 0.163 | 0.736 | 0.064 | 0.245 |
| 4 | 0.056 | 0.792 | 0.060 | 0.305 |
| 5 | 0.051 | 0.843 | 0.059 | 0.364 |
| 6 | 0.042 | 0.885 | 0.058 | 0.423 |
| 7 | 0.027 | 0.912 | 0.053 | 0.475 |
| 8 | 0.019 | 0.931 | 0.050 | 0.526 |

### 5.3. Fault Prediction

We randomly selected two-thirds of the data to train the model and used the remaining data as the test set to verify the predictive ability of the model. We selected accuracy, precision, recall, $F_1$ score, and AUC(Area Under Curve) as the evaluation indicators of the model's predictive ability and conducted 500 repeated experiments to obtain the average value of each indicator. Taking the hidden layer of BP neural network with 30 neurons as an example, the modeling time (500 repeated experiments) of CIPCA-BPNN and PCA-BPNN was 599.76 min and 301.86 min, respectively.

As can be seen from Figure 7a, CIPCA-BPNN could predict failures well. Its accuracy could reach more than 95% and precision was more than 90%; although the value of recall changed greatly, when the number of hidden neurons was more than 20, it could also reach 90%. In addition, the ensemble classifier effect of CIPCA-BPNN was better than the two base classifiers, indicating that the ensemble classifier used more data features, reflecting the advantages of CIPCA-BPNN. Finally, the model effect got better and better with the increase in the number of hidden neurons and tended to stabilize after the number of hidden neurons reached 30. In practical applications, more than 30 hidden neurons can be used to build a model, and a good prediction effect can be obtained.

We also constructed a PCA-BPNN prediction model and compared it with CIPCA-BPNN under the same circumstances. It can be seen from Figure 7b that the accuracy and precision of PCA-BPNN was slightly lower than that of CIPCA-BPNN and was much lower than CIPCA-BPNN in recall. In fault prediction, recall is more important than accuracy and precision because recall refers to the probability that a fault can be accurately predicted when it actually occurs. The AUC curve in Figure 7c also shows that the prediction effect of CIPCA-BPNN was better than that of PCA-BPNN. The details of each evaluation index of the model's prediction ability are shown in Table 3.

**Table 3.** Effects comparison of fault prediction by CIPCA-BPNN and PCA-BPNN.

| Number of Hidden Neurons | CIPCA-BPNN | | | | | PCA-BPNN | | | | |
|---|---|---|---|---|---|---|---|---|---|---|
| | Accuracy | Precision | Recall | $F_1$-Score | AUC | Accuracy | Precision | Recall | $F_1$-Score | AUC |
| 10 | 0.973 | 0.922 | 0.743 | 0.823 | 0.986 | 0.958 | 0.850 | 0.617 | 0.715 | 0.960 |
| 15 | 0.982 | 0.932 | 0.854 | 0.891 | 0.993 | 0.969 | 0.884 | 0.725 | 0.797 | 0.977 |
| 20 | 0.986 | 0.940 | 0.895 | 0.917 | 0.995 | 0.974 | 0.900 | 0.784 | 0.838 | 0.984 |
| 25 | 0.988 | 0.948 | 0.914 | 0.931 | 0.996 | 0.978 | 0.912 | 0.821 | 0.864 | 0.987 |
| 30 | 0.990 | 0.953 | 0.923 | 0.938 | 0.997 | 0.980 | 0.921 | 0.839 | 0.878 | 0.989 |
| 35 | 0.991 | 0.957 | 0.930 | 0.943 | 0.998 | 0.982 | 0.925 | 0.850 | 0.886 | 0.990 |
| 40 | 0.991 | 0.960 | 0.934 | 0.947 | 0.998 | 0.982 | 0.928 | 0.856 | 0.891 | 0.991 |
| 45 | 0.992 | 0.962 | 0.937 | 0.949 | 0.998 | 0.983 | 0.932 | 0.862 | 0.896 | 0.992 |

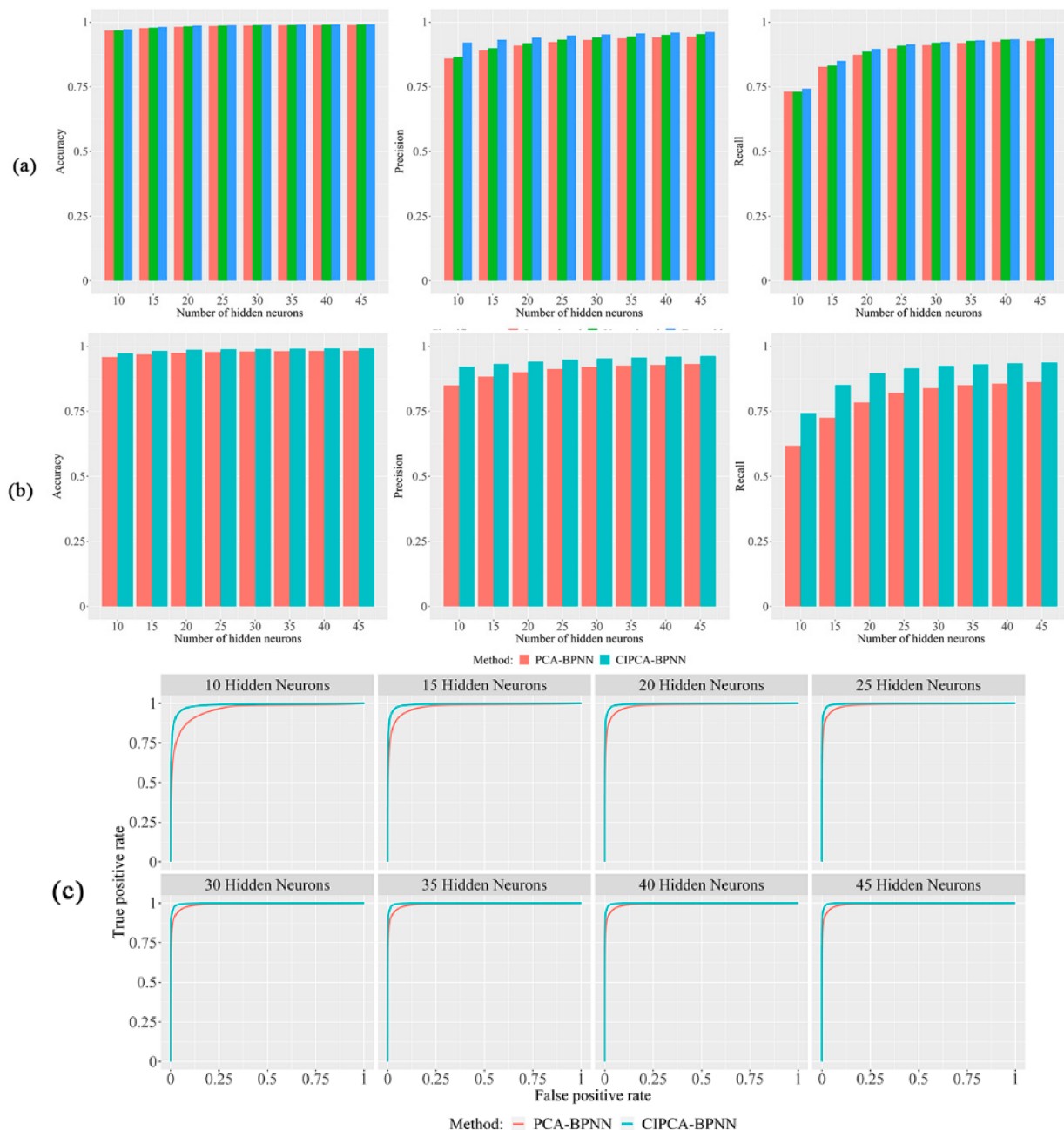

**Figure 7.** Experimental results and comparison. (**a**) Fault prediction effect on test data by CIPCA-BPNN. The "lower bond" is the result of lower bond based classifier; "upper bond" is the result of upper bond based classifiers; "ensemble" is the result of ensemble classifier. (**b**) Comparison of fault prediction effect on test data between PCA-BPNN and CIPCA-BPNN. "CIPCA-BPNN" is the result of CIPCA-BPNN's ensemble classifier. (**c**) Comparison of ROC(Receiver operating characteristic) curves on test data between PCA-BPNN and CIPCA-BPNN. "CIPCA-BPNN" is the result of CIPCA-BPNN's ensemble classifier.

In practice, if the fault can be accurately predicted before the fault occurs, it can help us avoid the fault or take measures to reduce the loss that the fault may cause. In order to achieve this goal, we calculated the CIPCA-BPNN test set prediction results at different times before the failure. Our experiment divided the flight data into 0.1 s intervals, but in practice, it is difficult to respond effectively to a fault with only a 0.1 s warning. Therefore, we summarized the prediction results within 1 s and observed the prediction effect of the model. For example, by summarizing the prediction results from 29 s to 30 s before the

fault occurs, we can know that the model can accurately predict the occurrence of the fault at 29 s.

We can see from Figure 8 that CIPCA-BPNN can predict the occurrence of a fault before it occurs. When modeling with 30 or more hidden neurons, we had two time windows that could predict in advance and accurately when a fault would occur. The first window was 16 s before the fault occurred. At this time, the accuracy, precision, and recall of the model were all close to 1, indicating that if the result of the model is "fault" at a particular time, there was a nearly 100% possibility of a fault 16 s later. But the first time window was very short, only 1 s. The second window was 9 s to 7 s before the fault occurred. In that window, we had 2 s to react to the fault that would occur. In addition, between 28 s and 2 s before the fault, the recall of CIPCA-BPNN was above 90%, and the accuracy was close to 1, so CIPCA-BPNN could fully predict the fault within 30 s before the fault occurred.

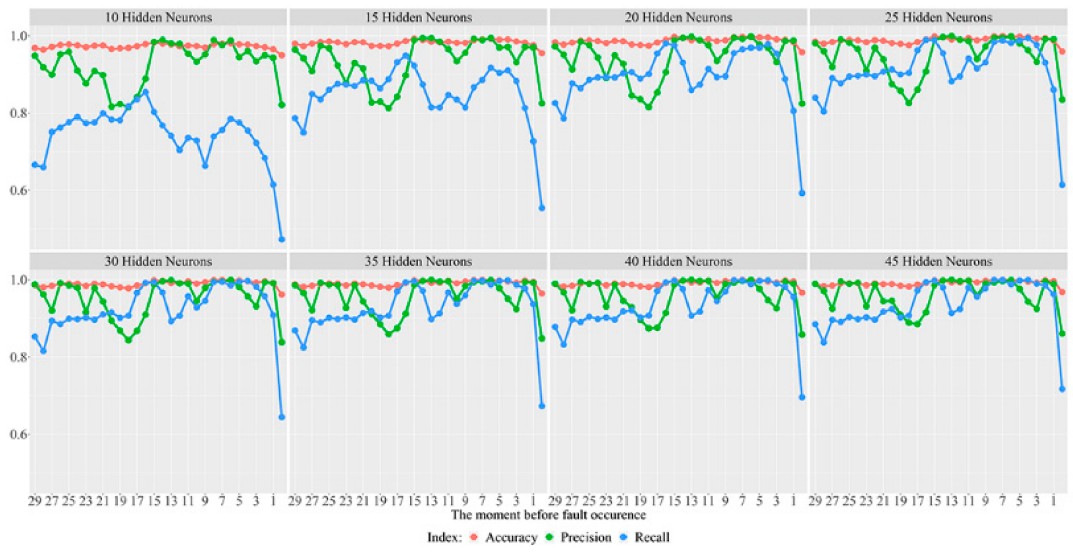

**Figure 8.** Failure prediction effect at each moment before failure by CIPCA-BPNN.

## 6. Conclusions

In this paper, we introduce the concept of interval data into fault prediction. Based on actual UAV flight data, a CIPCA-BPNN fault prediction model was established. CIPCA can achieve sample compression and dimensionality reduction of flight data on the basis of retaining the vast majority of data features and can reduce the data imbalance ratio to a certain extent. It can greatly shorten the modeling time of a fault prediction model, improving the modeling quality. By comparison, it has more advantages than the traditional PCA method. The experimental results show that the prediction effect of CIPCA-BPNN was better than the traditional PCA-BPNN model and could accurately predict the occurrence of a fault 9 to 7 s before the fault occurred. In the future, the model can be loaded into UAVs for practical application. The prediction results of the model can give the UVA operator precious time to deal with the failure before it happens, which has a strong practical significance.

**Author Contributions:** Conceptualization, L.Y. and S.Z.; data curation, L.Y.; formal analysis, L.Y.; investigation, L.Y.; methodology, L.Y. and C.L.; project administration, G.J., F.W., and S.Z.; resources, C.L.; software, L.Y.; supervision, L.Y., G.J., F.W., W.C., and S.Z.; validation, L.Y., G.J., F.W., and W.C.; visualization, L.Y.; writing—original draft, L.Y.; writing—review and editing, L.Y. All authors have read and agreed to the published version of the manuscript.

**Funding:** This research was supported by the National Natural Science Foundation of China (Grant No.71971013 & 71871003) and the Technical Research Foundation (Grant No.JSZL2016601A004). The study was also sponsored by the Fundamental Research Funds for the Central Universities (Grant No.YWF-20-BJ-J-943) and the Graduate Student Education & Development Foundation of Beihang University.

**Institutional Review Board Statement:** Not applicable.

**Informed Consent Statement:** Not applicable.

**Data Availability Statement:** Restrictions apply to the availability of these data. The data in this research came from Volitation (Beijing) Technology Co., Ltd. Please contact Linchao Yang (yanglinchao@buaa.edu.cn) to inform about the data availability.

**Acknowledgments:** All authors would like to thank the data support by Volitation (Beijing) Technology Co., Ltd.

**Conflicts of Interest:** The authors declare no conflict of interest. The funders had no role in the design of the study; in the collection, analyses, or interpretation of data; in the writing of the manuscript; or in the decision to publish the results.

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
