# Peer review of "The CIPCA-BPNN Failure Prediction Method Based on Interval Data Compression and Dimension Reduction"

_applsci, doi:10.3390/app11083448_

Round 1

Reviewer 1 Report

The authors proposed a new method for fault prediction based on CIPCA-BPNN. The paper is interesting. Comments and suggestions are as follow

1). The mistake of using a hyphen in line 36.

2). In the introduction section, it would be better to add the content of the difference between WPT, EMD and LMD.

3). It should be uk'ul=0 in line 123.

4). Please check the expression of 'flying ginseng system'. I think it is a wrong expression.

5). In the compression step, how to determine the available time span of the interval since the dimension is closely related to the time span?

6). How to obtain Fig. 5. Since it shows the correlation of the variables, the diagonal entries mean the autocorrelation coefficient. They all should be 1 but they are not in Fig. 5.

7). In table 1, row 2 and row 3 are the same, it would be better to delete one of them.

Author Response

Dear Reviewer,

On behalf of my co-authors, we thank you very much for giving us an opportunity to revise our manuscript. we appreciate you very much for the positive and constructive comments and suggestions on our manuscript entitled “The CIPCA-BPNN Failure Prediction Method Based on Interval Data Compression and Dimension Reduction”. (Manuscript ID: applsci-1161380).

We have studied reviewer’s comments carefully and have made revision in the paper. The main corrections in the paper and the responds to the reviewer’s comments are as following:

Point 1: The mistake of using a hyphen in line 36.

Response 1: Thank you for your comments. This suggestion is very necessary. We have changed "sup-port" to "support" in line 36.

Point 2: In the introduction section, it would be better to add the content of the difference between WPT, EMD and LMD.

Response 2: Thank you for your comments. According to your suggestion, we have compared the three methods and added the following contents in line 62: “WPT needs to determine the decomposition scale, so it is not an adaptive signal data processing method, and is not conducive to processing big data. EMD can adaptively determine the resolution of the signal in different frequency bands, but the modal mixing problem often occurs. LMD and EMD have some similarities, but LMD is better than EMD in the processing of local signal features.”

Point 3: It should be uk'ul=0 in line 123.

Response 3: Thank you for your comments. We have corrected this formula to “” in line 129.

Point 4: Please check the expression of 'flying ginseng system'. I think it is a wrong expression.

Response 4: Thank you for your comments. We are sorry that this has caused you trouble. We want to show that the flight data collected by a large number of airborne sensors of UAV has the characteristics of multivariability. We have changed the sentence to “Second, there are a large number of flight status indicators in flight data, ranging from dozens to hundreds.” in line 180.

Point 5: In the compression step, how to determine the available time span of the interval since the dimension is closely related to the time span?

Response 5: Thank you for your comments. Due to different data characteristics and purposes of data analysis, researchers should decide the time span of the interval according to their own needs. However, we have emphasized in line 194 that the time span of each interval should be the same to preserve the time characteristics of the data. Of course, the choice of the optimal time span is also one of our ongoing studies.

Point 6: How to obtain Fig. 5. Since it shows the correlation of the variables, the diagonal entries mean the autocorrelation coefficient. They all should be 1 but they are not in Fig. 5.

Response 6: Thank you for your comments. Since there are many variables and the autocorrelation coefficient of all variables is 1, we did not show the autocorrelation coefficient in Figure 5. The diagonal line in Figure 5 shows the correlation coefficients of two adjacent variables (the one to the left and the one above the grid). We have included the caption in line 258.

Point 7: In table 1, row 2 and row 3 are the same, it would be better to delete one of them.

Response 7: Thank you for your comments. The row 3 in Table 1 is intended to illustrate what happens to the data after dimensional reduction. We have modified row 2, column 6 in Table 1.

Sincerely yours,

Linchao Yang on behalf of the authors.

Reviewer 2 Report

The paper presents interesting approach to reducing data and fault prediction. The description of NN in 2.3 should have some references to fundamental books about NN and learning algorithms. Also information about learning algorithm should be included.
Conclusions section should contained more information about advantages of the presented approach.

Because the experimental data are available it is important for the reader to provide also computational time and computational resources (processors, FPGA) needed for real time data processing. Is this approached planned to be used as on-UAV compression, data transfer, fault analysis on some remote “base” or in-flight on-UAV processing and fault prediction?

Author Response

Dear Reviewer,

On behalf of my co-authors, we thank you very much for giving us an opportunity to revise our manuscript. we appreciate you very much for the positive and constructive comments and suggestions on our manuscript entitled “The CIPCA-BPNN Failure Prediction Method Based on Interval Data Compression and Dimension Reduction”. (Manuscript ID: applsci-1161380).

We have studied reviewer’s comments carefully and have made revision in the paper. The main corrections in the paper and the responds to the reviewer’s comments are as following:

Point 1: The description of NN in 2.3 should have some references to fundamental books about NN and learning algorithms. Also information about learning algorithm should be included.

Response 1:Thank you very much for your comments. According to your suggestions, we have added classic books such as Advanced Methods in Neural Computing and Advanced Methods in Neural Computing to the references ([33-35]). In addition, we have explained the relationship between BP algorithm and neural network in line 144.

Point 2: Conclusions section should contained more information about advantages of the presented approach.

Response 2: Thank you for your comments. This suggestion is very necessary. CIPCA can achieve sample compression and dimensionality reduction of flight data on the basis of retaining the vast majority of data features, and reduce the data imbalance ratio to a certain extent. It can greatly shorten the modeling time of fault prediction model and improve the modeling quality. By comparison, it has more advantages than traditional PCA method. Moreover, CIPCA-BPNN is better than PCA-BPNN in fault prediction. According to your suggestions, we have modified the conclusion and added the relevant contents about the advantages of CIPCA-BPNN in data preprocessing and feature extraction as well as fault prediction in line 391.

Point 3: Because the experimental data are available it is important for the reader to provide also computational time and computational resources (processors, FPGA) needed for real time data processing.

Response 3: Thank you for your comments. The computational resources of this study are AMD Ryzen 7 1700 Eight-Core Processor 3 GHz CPU and 32 GB RAM. We have added a description of the computational resources to line 289. Taking 30 neurons in the hidden layer as an example, the modeling time (500 repeated experiments) of CIPCA-BPNN and PCA-BPNN is 599.76 minutes and 301.86 minutes respectively. We have added a description of the computational time to line 334.

Point 4: Is this approached planned to be used as on-UAV compression, data transfer, fault analysis on some remote “base” or in-flight on-UAV processing and fault prediction?

Response 4: Thank you for your comments. Our goal is to apply the fault prediction model to the real-time fault prediction of UAV in flight. But field tests are needed. In the future, it could be used in the practical work of UAVs. We have added that to line 397.

Sincerely yours,

Linchao Yang on behalf of the authors.
